# Mie-coupled bound guided states in nanowire geometric superlattices

Seokhyoung Kim [1], Kyoung-Ho Kim[1,2], David J. Hill [1], Hong-Gyu Park[2] & James F. Cahoon [1]

All-optical operation holds promise as the future of computing technology, and key components include miniaturized waveguides (WGs) and couplers that control narrow bandwidths. Nanowires (NWs) offer an ideal platform for nanoscale WGs, but their utility has been limited by the lack of a comprehensive coupling scheme with band selectivity. Here, we introduce a NW geometric superlattice (GSL) that allows narrow-band guiding in Si NWs through coupling of a Mie resonance with a bound-guided state (BGS). Periodic diameter modulation creates a Mie-BGS-coupled excitation that manifests as a scattering dark state with a pronounced scattering dip in the Mie resonance. The frequency of the coupled mode, tunable from the visible to near-infrared, is determined by the pitch of the GSL. Using a combined GSL-WG system, we demonstrate spectrally selective guiding and optical switching and sensing at telecommunication wavelengths, highlighting the potential to use NW GSLs for the design of on-chip optical components.

[1] Department of Chemistry, University of North Carolina-Chapel Hill, Chapel Hill, NC 27599-3290, USA. [2] Department of Physics and KU-KIST Graduate School of Converging Science and Technology, Korea University, Seoul 02841, Republic of Korea. Correspondence and requests for materials should be addressed to H.-G.P. (email: hgpark@korea.ac.kr) or to J.F.C. (email: jfcahoon@unc.edu)

Optical interconnects and logic elements operating on the nanometer scale could enable the leap into all-optical computing technology, which would alleviate the bandwidth and energy consumption limitations of current technologies[1]. For instance, dielectric waveguides (WGs) made on silicon-on-insulator (SOI) platforms have been widely explored for carrying and modulating optical signals[2]; however, they are confined to a planar geometry, and coupling light into and out of these structures relies on conventional grating couplers that are often orders of magnitude larger in size than the WGs[3,4]. Nanowires (NWs)[5] offer an alternate platform on which to design dielectric WGs at exactly the diffraction limit in an optimal cylindrical geometry[6–9]. NWs enable the guiding of light in flexible, non-planar, three-dimensional geometries, and permit the addition of conformal cladding materials including dielectrics and metals[10]. Despite these advantages, in-coupling of light to NW WGs remains problematic and has relied on scattering at NW end facets[6,7] or end-on parallel coupling to the NWs[8,9] because there is no comprehensive light coupling scheme for a NW geometry. Moreover, existing methods are not spectrally selective, limiting their utility for applications where mode-selection is important. Given that NWs are also well known to exhibit strong Mie resonances[11,12], there have been efforts to explore the connection between Mie resonances and guided modes[13]. However, because of momentum mismatch, guided modes are by their nature inaccessible optical bound states[14] under normal incidence light, leaving the opportunities for interplay between a Mie and guided mode as yet obscure.

Nanoscale systems often exhibit unique light-matter interactions[15] when the geometric and compositional design of a system causes complex wave interference effects, which can include Fano resonances[16], electromagnetically induced transparency (EIT)[17], and scattering dark states[18,19]. A Fano resonance arises from weak coupling between two resonances with different damping rates, creating an abrupt spectral feature in the optical response. EIT and scattering dark states originate from oppositely-oriented radiative resonances that destructively interfere with each other in the far field[17–21]. The same optical effect can occur in systems with a near-field interaction between a super-radiant, bright mode and a sub-radiant, dark mode[22,23]. In this case, an exchange of energy between the modes results in a coupled excitation of the bright and dark state, causing an abrupt Fano-like or EIT-like feature in the optical response. Far-field and near-field interactions are usually interchangeable frameworks that can describe the same phenomena with a different basis set[18,24]. These spectral features are often generated through geometric perturbation of a nanostructure to break spatial symmetry, causing a previously dark state to be accessed[17,22,25,26].

Here, we show that a NW geometric superlattice (GSL), which breaks the infinite translational symmetry of a NW, can couple a Mie resonance with a bound-guided state (BGS) of the NW. Complementary to reports on bound states in the continuum (BIC)[14], which describe the appearance of bound states above the light cone, we selectively access one normally inaccessible BGS from a continuum of conventional bound states below the light cone by precisely modulating the NW geometry. As a result, a sharp, Fano-like feature that can be assigned as a scattering dark state appears in the Mie scattering spectrum at the wavelength of the BGS. The geometric dependence of both the Mie resonance and BGS provides broadband tuning of the effect from visible through near-infrared wavelengths. We experimentally demonstrate the appearance of scattering dark states from NW GSLs and show selective guiding of light up to telecommunication wavelengths with a Fourier-transform-limited bandwidth. This report is the first demonstration of Mie-BGS coupling and selective narrow-band guiding in a NW WG system. In addition, we show that the coupling wavelength is highly sensitive to the local

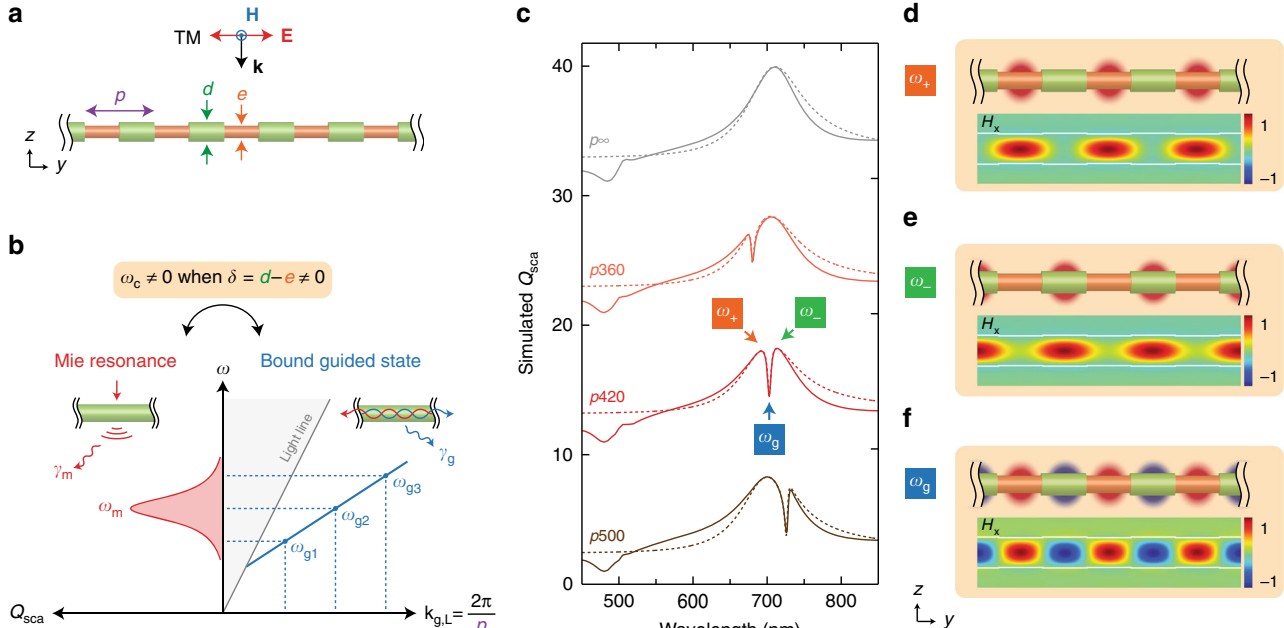

**Fig. 1** The geometry and coupled resonances of a NW GSL. **a** Structure and illumination geometry of a Si NW GSL. **b** Schematic illustration of a leaky Mie resonance with a broad spectral envelope (left) and a continuum of BGS states (right), in which a specific BGS state, $\omega_g$, is selected by the choice of pitch $p$. **c** Numerically calculated spectra of $Q_{sca}$ (solid lines) and analytical fit by TCMT (dotted lines) for an infinite NW with uniform $d$ ($\delta = 0$) of 140 nm (gray) and for NW GSLs with $d = 140$ nm, $e = 135$ nm, and varying $p$ of 360 nm (orange), 420 nm (red), and 500 nm (brown). The pitch, $p$, in nanometers is denoted for each spectrum, where $p = \infty$ corresponds to an infinite NW of uniform diameter $d$. All spectra are successively offset vertically by 10. **d–f** Schematic and calculated magnetic field profiles for a GSL with $p = 420$ nm at frequencies of $\omega_+$ (**d**), $\omega_-$ (**e**), and $\omega_g$ (**f**), corresponding to the red spectrum and labels in **c** at wavelengths of 692, 713, and 703 nm, respectively

refractive index, an effect that we use to design a sensor and optical switch.

## Results

**Mie-BGS coupling in a NW GSL.** The geometry of a NW GSL, illustrated in Fig. 1a, is uniquely defined by three geometrical parameters, the pitch ($p$), outer diameter ($d$), and etch diameter ($e$), and is composed of equally spaced cylindrical segments of modulated diameter ($\delta = d - e$), where each segment has an axial length of $p/2$. We consider this structure illuminated at normal incidence under TM polarization with the magnetic (**H**) and electric (**E**) fields perpendicular and parallel, respectively, to the NW axis. Figure 1b schematically illustrates coupling between the two eigenmodes in a NW geometry, a Mie resonance that strongly couples with a free-space plane wave and a BGS selected from the continuum. Here, we consider the criterion for an ideal bound state to be absence of loss, absence of external coupling, and time invariance of energy within the structure. Thus, for a NW system, the ideal BGS refers to a standing wave with amplitude $A_g$ formed by the superposition of two counter-propagating guided modes, $A_g = [A_r(\mathbf{x,z})B(\mathbf{y})\exp(i\mathbf{k}_{g,L}(\mathbf{y} - \mathbf{y_0}) - i\omega_g t) + A_l(\mathbf{x,z})B(\mathbf{y})\exp(-i\mathbf{k}_{g,L}(\mathbf{y} - \mathbf{y_0}) - i\omega_g t)]$, with the same angular frequency ($\omega_g$) and longitudinal wavevector ($\mathbf{k}_{g,L}$), where $\mathbf{y}$ is spatial position along the NW axis, $\mathbf{y_0}$ is the spatial origin chosen to coincide with the center of a GSL segment, $A_r(\mathbf{x, z})$ and $A_l(\mathbf{x,z})$ are the cross-sectional amplitudes of right- and left-propagating guided modes, respectively, (with $A_r = A_l$, for an ideal BGS), $B(\mathbf{y})$ is the longitudinal envelope function (with $B(\mathbf{y})$ = constant, for an ideal BGS), i is the imaginary unit ($i^2 = -1$), and $t$ is time. Note that this expression will also describe a conventional guided mode if either $A_r$ or $A_l$ is zero. For a perfect cylinder ($\delta = 0$), the Mie and BGS modes are orthogonal and incapable of coupling to one another. For a GSL ($\delta > 0$), we consider a non-zero coupling frequency, $\omega_c$, between a Mie resonance centered at $\omega_m$ and a BGS centered at $\omega_g$. For normal incidence illumination, the coupling is non-zero only for the $\omega_g$ with a $\mathbf{k}_{g,L}$ that matches the geometrically defined $p$ (i.e., $\mathbf{k}_{g,L} = 2\pi m/p$, where $m = 1, 3, 5, \ldots$) because these frequencies experience coherent amplification whereas mismatched ones vanish. Thus, the structure behaves as a Fourier selector in which only the spatial frequency components, including higher $m$th order odd harmonics (Supplementary Fig. 1), matched to the structure are coupled to a BGS.

Figure 1c shows numerically calculated scattering efficiency ($Q_{sca}$, see Methods for details) spectra (solid lines) and analytically calculated spectra (dashed lines) derived from temporal coupled-mode theory (TCMT)[18,27] for a uniform NW (labeled $p\infty$) and three NW GSLs of varying $p$. The dominant peak in the uniform NW spectrum corresponds to the fundamental $TM_{11}$ magnetic dipolar Mie resonance (directly related to the $HE_{11}$ guided mode[13]) centered at $\omega_m$. For the GSL, in contrast, a sharp and pronounced scattering dip (labeled $\omega_g$ for $p420$) is observed within the Mie resonance. The calculated field profile at $\omega_g$ (Fig. 1f) exhibits a standing wave with antinodes centered on each cylindrical segment (see also a time-lapsed animation in Supplementary Movie 1). This standing wave profile corresponds to the BGS. For shorter ($p360$) and longer ($p500$) values of $p$, the scattering dip shifts to the blue or red of $\omega_m$, respectively; nevertheless, the same standing wave profile is observed at the scattering dip for all values of $p$ (Supplementary Fig. 2). The peaks to the blue and red of the scattering dip in Fig. 1c (labeled $\omega_+$ and $\omega_-$, respectively, for $p420$) exhibit mode profiles (Fig. 1d, e) with antinodes centered only on the small or large diameter segments, respectively, and these profiles are also preserved for every value of $p$ (Supplementary Fig. 2).

Substantially larger values of $p$ shift $\omega_g$ off the Mie resonance envelope, causing the scattering dip to disappear and no power to be transferred to the BGS (Supplementary Fig. 3). In addition, a tilt of the incoming plane wave provides a non-zero momentum along the NW axis, causing the BGS scattering dip to split into two separate dips dominated by a right- and left-propagating BGS state (Supplementary Fig. 4); however, we focus only on normal incidence illumination in this study.

The features of the Mie–BGS coupling can be accounted for using TCMT[18,27]. The dynamic state of the resonance amplitudes can be expressed as:

$$\frac{d}{dt}\begin{pmatrix} A_m \\ A_g \end{pmatrix} = \left[ -i\begin{pmatrix} \omega_m & \omega_c \\ \omega_c & \omega_g \end{pmatrix} - \begin{pmatrix} \gamma_m & 0 \\ 0 & \gamma_g \end{pmatrix} \right]\begin{pmatrix} A_m \\ A_g \end{pmatrix} + \begin{pmatrix} k_m \\ 0 \end{pmatrix}S_+, \tag{1}$$

where $A_m$ is the Mie resonance amplitude, $\gamma_m$ and $\gamma_g$ are radiative decay rates for the Mie resonance and BGS, $S_+$ is the incoming wave, and $k_m$ is the coupling coefficient to the incoming wave. Here, we have neglected absorptive contributions and neglected an off-diagonal radiative coupling term, $\gamma_c$. The outgoing wave, $S_-$, can be expressed as $S_- = S_+ + d_m A_m$, where $d_m$ is the coupling coefficient to the outgoing wave. The coupling coefficients of the BGS to the incoming and outgoing waves are set to zero under normal incidence illumination, which assumes that $\gamma_g$ is small compared to $\gamma_m$. In a perfectly cylindrical NW geometry ($\delta = 0$), coupling between the Mie resonance and BGS does not occur, and $\omega_c$ and $\gamma_g$ are both zero, leaving both resonances unaffected and the BGS completely dark. However, in the case of a NW GSL, both parameters are non-zero, and scattering cross-sections calculated using Eq. (1) are plotted in Fig. 1c as dotted lines. Analytical parameters are summarized in Supplementary Table 1 and were determined by a fit to the numerical simulation considering energy conservation and time reversal symmetry[18], reproducing the spectral features in the numerical results. Note that the scattering dip does not go to zero because of the introduction of loss in the guided mode ($\gamma_g > 0$; see Supplementary Fig. 5). The good agreement of numerical and TCMT results validates the assignment of the scattering dip as a scattering dark state resulting from coupling between a Mie resonance and BGS. Similar calculations with different geometrical parameters (Supplementary Fig. 6) show that the effect is fully tunable from visible through infrared wavelengths by changing $d$ and $p$ with a fixed $\delta$.

Diagonalization of the energy matrix in Eq. (1) yields eigenfrequencies, $\omega_+$ and $\omega_-$, and the eigenmodes, $A_+$ and $A_-$. If the scattering dip appears at the center of the Mie resonance ($\omega_m = \omega_g$; red curve in Fig. 1c), the eigenmodes are $A_\pm = A_m \pm A_g$, where $A_m$ corresponds to a dipolar Mie resonance uniformly distributed across the axial length of the NW, and $A_g$ is the standing-wave mode profile of the BGS. The eigenmodes $A_+$ and $A_-$ thus represent interference between the Mie resonance and BGS, in which the axially uniform amplitude of the Mie resonance overpowers the oscillating amplitude of the BGS with a relative phase shift of $\pi$ for the two eigenmodes. This result corresponds to the numerically calculated mode profiles in Fig. 1d, e for the adjacent peaks on the blue ($\omega_+$) and red ($\omega_-$) sides of the scattering dip. This eigenmode analysis can also justify the assignment of the scattering dip as a Fano resonance arising from the interaction between a sharp BGS and a broad Mie background resonance.

The bound character of the BGS is apparent if we consider the effect of the modulation depth, $\delta$. Numerical simulations (Supplementary Fig. 7) with $\delta$ ranging from 20 to 2 nm show a

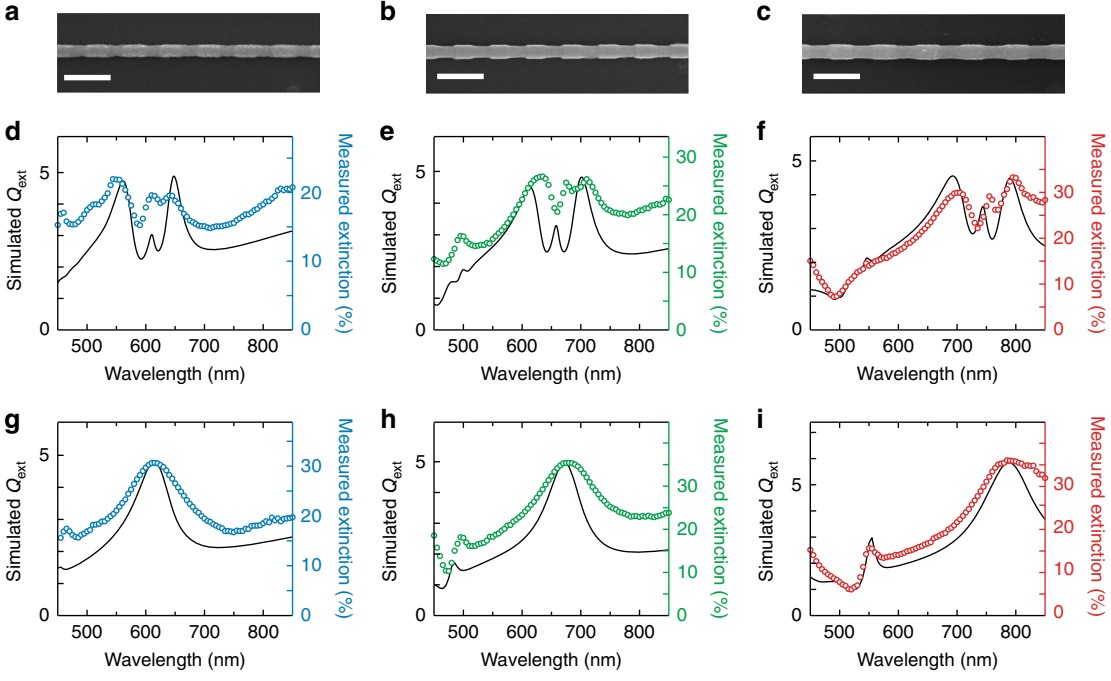

**Fig. 2** Experimental extinction spectra of NW GSLs. **a–c** SEM images of three NW GSL with geometrical parameters of $d = 130 \pm 1$ nm, $e = 114 \pm 1$ nm, $p = 397 \pm 10$ nm (**a**), $d = 145 \pm 1$ nm, $e = 119 \pm 5$ nm, $p = 383 \pm 9$ nm (**b**), and $d = 158 \pm 1$ nm, $e = 134 \pm 2$ nm, $p = 465 \pm 10$ nm (**c**); scale bars, 500 nm. **d–i** Extinction spectra of NW GSLs (**d–f**) and of uniform NWs of the same diameters (**g–i**), showing experimentally measured extinction (colored circles and right-hand axes) and numerically simulated $Q_{\text{ext}}$ (black lines and left-hand axes). Spectra in **d–f** correspond to the images in **a–c**, respectively

marked narrowing of the scattering dark state linewidth and increase of the quality factor, extrapolating to infinity as $\delta$ approaches zero. Moreover, $\omega_c$ decreases as $\delta$ decreases in magnitude (Supplementary Table 2) because access to the bound state vanishes. In contrast, as $\delta$ increases in magnitude, the linewidth significantly broadens and eventually bifurcates into two distinct peaks that align with the positions of $\omega_+$ and $\omega_-$. This transition occurs because the Mie resonance and BGS are no longer weakly coupled, and instead two qualitatively new modes arise that are more representative of scattering from Si NW segments of distinct diameters.

To experimentally verify the predicted optical properties of NW GSLs, a process termed ENGRAVE (Encoded Nanowire GRowth and Appearance through VLS and Etching)[28–30] was used to fabricate Si NW GSL nanostructures from the bottom-up using a gold (Au) nanoparticle catalyzed vapor-liquid-solid (VLS) growth process. VLS growth produces single-crystalline NWs and is capable of precise size control and on-demand modification of morphology[28,29,31–33], chemical composition[34], and crystal phase[35]. Of the geometric parameters, $p$ is determined by the dopant switching frequency during the growth, and $d$ and $e$ are determined by the size of Au catalyst and post-growth etch duration, respectively. Optimized growth conditions without Au loss from the catalyst[36] yield NWs exceeding 500 µm in length with no undesired diameter variation (Supplementary Fig. 8). Figure 2a–c shows scanning electron microscope (SEM) images of three distinct NW GSLs with increasing diameters $d$ prepared by the ENGRAVE process using etch depths of $\delta = 10$–25 nm. Geometrical parameters of each GSL were determined by quantitative image analysis[30]. For optical characterization, a home-built laser microscope (Supplementary Fig. 9) capable of polarization-resolved bright-field extinction measurements using a supercontinuum laser source and balanced detection was employed.

Experimental extinction spectra, plotted as $(1 - T) \times 100\%$, where $T$ is the measured transmittance (see Methods), are shown

in Fig. 2d–i for GSLs (Fig. 2d–f) and uniform NWs of the same diameters (Fig. 2g–i). Each spectrum shows a broad Mie resonance that shifts to longer wavelengths with increasing diameter. For the GSLs, sharp spectral features deviating from the background Mie resonances are observed. Numerical simulations (black solid lines) of the extinction efficiency ($Q_{\text{ext}}$) that use geometrical parameters obtained from the SEM images and include the focused laser beam (see Methods) agree well with the experimental measurements, confirming the presence of Mie-BGS coupling and a scattering dark state in these nanostructures. However, the simulations also reveal several differences between focused excitation and the plane wave excitation modeled in Fig. 1. First, the simulations predict a scattering dip with a substantially larger linewidth, which can be attributed to the localized excitation causing a broader distribution of Fourier spatial components to couple to the BGS. Second, the simulations predict an extinction peak within the dip, which can be attributed to mode distortion caused by the finite number of GSL periods excited by the focused beam. Mode degradation in truncated arrays is a common phenomenon observed in planar metamaterials due to loss of coherence[37,38]. In our case, the finite beam prevents the Bloch condition along the GSL from being satisfied, causing incomplete formation of the BGS and an increase in scattering loss, manifesting as a peak within the scattering dip (Supplementary Fig. 10).

**Tunable guiding in a GSL-WG NW**. The geometric dependence of Mie-BGS coupling, as observed in Fig. 2, can be predicted from the dispersion relation for the WG modes of a uniform diameter Si NW (Fig. 3a) and the diameter dependence of the $TM_{11}$ Mie resonance. For a fixed longitudinal wavevector $\mathbf{k}_{\text{g,L}}$, which corresponds to a GSL satisfying $p = 2\pi m/\mathbf{k}_{\text{g,L}}$, the frequency of the guided mode decreases with increasing diameter, and the Mie resonance frequency similarly decreases with increasing diameter. However, because the frequencies of the Mie resonance and BGS

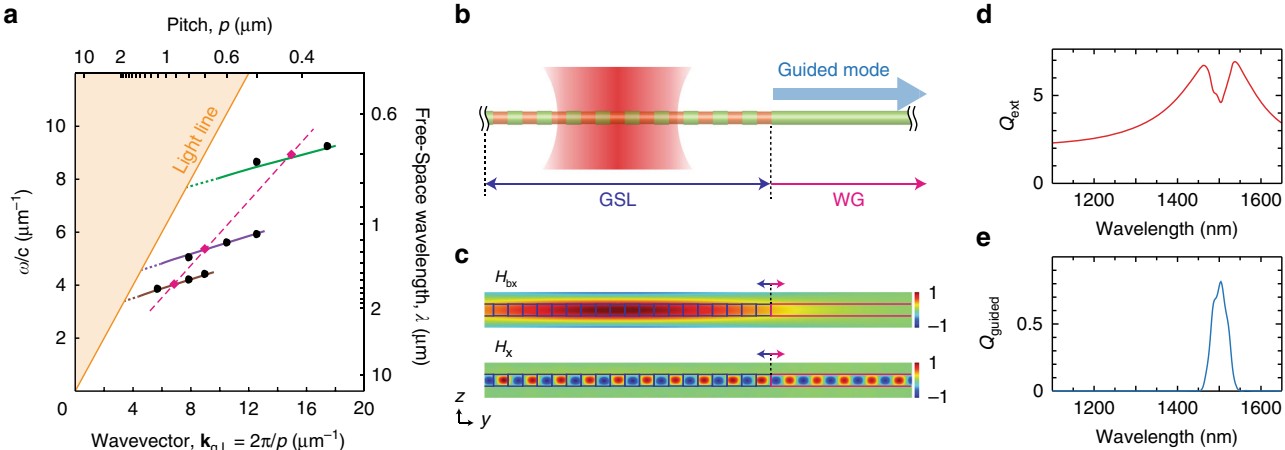

**Fig. 3** Selective excitation of guided modes through a NW GSL. **a** Dispersion curves (solid lines) of fundamental $HE_{11}$ guided modes in infinitely long NWs with $d = 140$ nm (green), $d = 250$ nm (purple), and $d = 340$ nm (brown). Marker symbols denote the positions of scattering dark states that are centered on a Mie resonance (magenta diamonds) or off-center from a Mie resonance (black circles). Magenta dashed line represents the frequency of a BGS that is centered on a Mie resonance for each value of $d$, so the intersection of solid and dashed lines represents the condition $\omega_m = \omega_g$. **b** Structure and excitation geometry of a NW GSL connected to a NW WG. **c** Calculated magnetic field profiles of an incoming Gaussian beam background field, $H_{bx}$, (top) and an excited mode scattered field, $H_x$, (bottom) of a NW GSL with $d = 330$ nm, $e = 310$ nm and $p = 900$ nm, and 10 periods, showing propagation from the GSL into the WG. **d, e** Simulated $Q_{ext}$ (**d**) and $Q_{guided}$ (**e**) spectra corresponding to the structure in **b** and **c**

change with diameter at different rates, the pitch $p$ required to center the BGS on the Mie resonance shifts to larger values as the diameter of the NW increases and the frequency decreases. In Fig. 3a, this shift can be visualized as the intersection of the solid lines (BGS dispersion relation) with the magenta dashed line (Mie resonance position), which represents the degenerate point where $\omega_m = \omega_g$ and provides a guideline for the geometric design of a GSL to enable optimal Mie-BGS coupling at a given free-space wavelength. Magenta diamonds and black circles represent data points where the modes are and are not degenerate, respectively, as determined from numerical simulations of NW GSL spectra in Fig. 1c and Supplementary Fig. 6. The good alignment of these data points with the dispersion relations confirms the geometric relationship between $d$ and $p$ for Mie-BGS coupling.

To take advantage of the WG characteristics, we designed a GSL-WG NW, illustrated in Fig. 3b, composed of a truncated GSL adjacent to a long, uniform cylindrical segment that can act as a lossless WG. The geometric parameters of the GSL were designed to support Mie-BGS coupling in the near-infrared spectral region to avoid absorptive losses in Si. Numerical simulations for a GSL ($d = 330$ nm; $e = 310$ nm; $p = 900$ nm) with 10 periods of the pitch illuminated by a focused laser beam (Fig. 3c) yield an extinction spectrum (Fig. 3d) with a broad Mie resonance centered at ~1500 nm and an extinction dip centered at ~1504 nm. A Mie-BGS coupled excitation occurs at the dip, and the numerical simulations show energy propagation into the guided mode of the WG segment adjacent to the GSL, as shown by the magnetic field profile in Fig. 3c and Supplementary Movie 2. Integration of the energy flux in the WG portion of the NW yields the guided spectrum shown in Fig. 3e. As expected, the spectral region guided in the NW corresponds to the spectral region of the dip in the extinction spectrum. The guided bandwidth of ~46 nm is in good agreement with the expected Fourier-transform-limited bandwidth of 48 nm for a Gaussian beam with a full-width at half maximum of 7 μm. Moreover, we calculate a maximum guided efficiency ($Q_{guided}$) of 0.82 for coupling into the WG based on the ratio of the power measured at the end of the WG to the power incident on the projected area of the GSL. Thus, the near-unity $Q_{guided}$

highlights the high efficiency with which the Mie-BGS coupling mechanism can funnel the energy associated with the strong light-matter interaction of the Mie resonance into the guided mode of a NW.

To experimentally validate the expected guiding characteristics of a GSL-WG NW, we synthesized NWs with a GSL adjacent to a 50 μm WG segment, as illustrated in Fig. 4a. Geometric parameters of the NW were chosen to support lossless guiding at frequencies below the bandgap of Si. Because of infrared free carrier absorption losses in heavily-doped n-type Si NWs[39], we adopted the doping-inverted ENGRAVE process[29,36] in which n-type rather than intrinsic segments are etched (see Methods), allowing the WG segment to be composed of intrinsic Si. To experimentally measure guiding in the GSL-WG structure, an excitation beam was focused on the GSL segment, and optical images were collected to observe any light emission from the end of the WG segment, where light is expected to scatter after traversing the 50 μm WG. Figure 4c–e displays optical images of a GSL-WG structure illuminated at three excitation wavelengths. As apparent from these images, WG light emission is observed from the end of the WG segment for an excitation wavelength of 1455 nm but is not observed at wavelengths of 1400 and 1550 nm.

Extinction spectra were collected from both GSL and WG segments of NWs with three distinct geometric parameters (Fig. 4i–n) designed to tune the Mie-BGS coupling from ~1300 to ~1500 nm. As expected, the uniform-diameter WG segments show one Mie resonance whereas the GSL segments show a scattering dark state within the Mie resonance. WG emission spectra, as shown in Fig. 4o–q, were also experimentally measured by integrating the signal from the end of the WGs (e.g., from the region denoted by a dashed circle in Fig. 4b). The WG emission spectra show that guiding occurs over a select wavelength range that corresponds to the spectral position of the scattering dip in the extinction spectra. The WG spectrum in Fig. 4o was collected from the same NW imaged in Fig. 4b, and the spectrum shows maximal WG intensity in the 1450–1500 nm range, in agreement with the images in Fig. 4c–e. Simulations of $Q_{ext}$ and $Q_{guided}$ spectra are shown as solid black lines in Fig. 4i–q and agree well with experimental measurements, confirming the Fourier-selected guiding characteristics of the GSL-WG NW

system. $Q_{guided}$ ranges from 0.5 to 1.0 depending on the choice of geometrical parameters.

Interestingly, the guiding spectra exhibit a high frequency oscillation (Fig. 4o–q), which indicates that the WG acts as a Fabry-Perot cavity. To confirm this effect, we prepared a single GSL with WG segments of two different lengths on either side

(Supplementary Fig. 11). WG emission was observed from the ends of both WGs, and spectra collected from each end (Fig. 4r–s) show free spectral ranges (FSRs) of 16.9 and 7.1 nm as well as finesse values of 1.58 and 1.17 from $WG_1$ and $WG_2$, respectively. The FSRs are in good agreement with the expected values of ~15 and ~6 nm based on respective WG cavity lengths, and the finesse

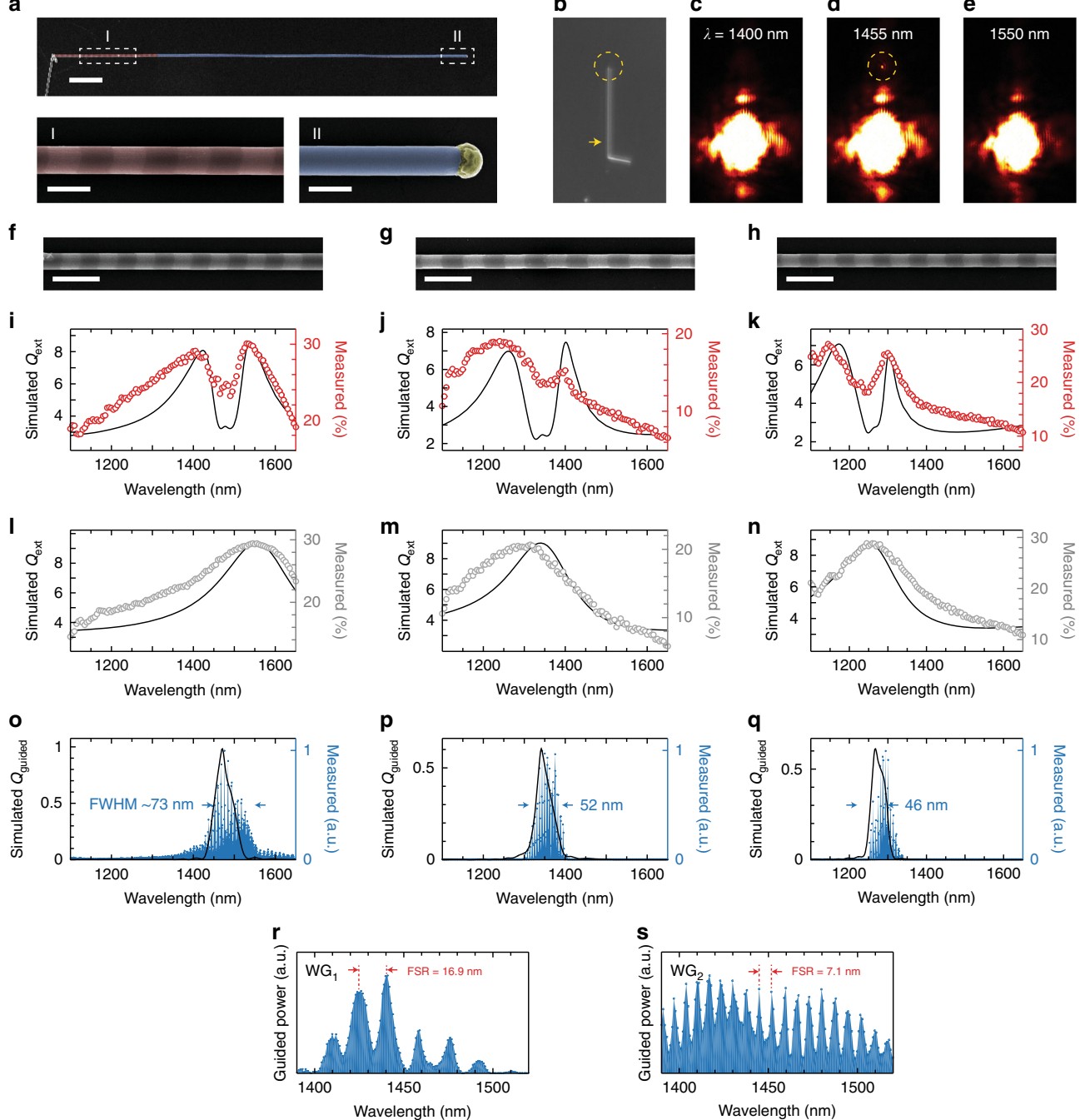

**Fig. 4** Experimental validation of tunable, narrow-band guiding by NW GSLs. **a** (Top) False-colored SEM image of a NW GSL (red) with $d = 334 \pm 1$ nm and $p = 781 \pm 9$ nm prepared by the doping-inverted ENGRAVE process followed by a 50 μm-long NW WG (blue); scale bar: 5 μm. (Bottom) Magnified views of dotted regions in the top image; scale bars: 500 nm. **b–e** False-colored optical images of the NW from **a** without an excitation beam (**b**) and under excitation at wavelengths of 1400 nm (**c**), 1455 nm (**d**), and 1550 nm (**e**). Yellow arrow and circles denote locations of the excitation beam and the NW tip, respectively. **f–h** SEM images of GSLs with geometrical parameters of $d = 334 \pm 1$ nm, $p = 781 \pm 9$ nm (**f**), $d = 284 \pm 3$ nm, $p = 882 \pm 6$ nm (**g**), $d = 266 \pm 1$ nm, $p = 772 \pm 6$ nm (**h**); scale bars, 1 μm. **i–q** Spectra of GSL extinction (**i–k**), NW WG extinction (**l–n**), and WG emission (**o–q**). Experimental measurements (right-hand axes) are shown as circles (and blue shaded area for guided spectra), and simulated spectra (left-hand axes) are shown as black lines. **r–s** Fabry-Perot patterns in guided power spectra for NW WGs with length of 21 μm (**r**, $WG_1$) and 53 μm (**s**, $WG_2$)

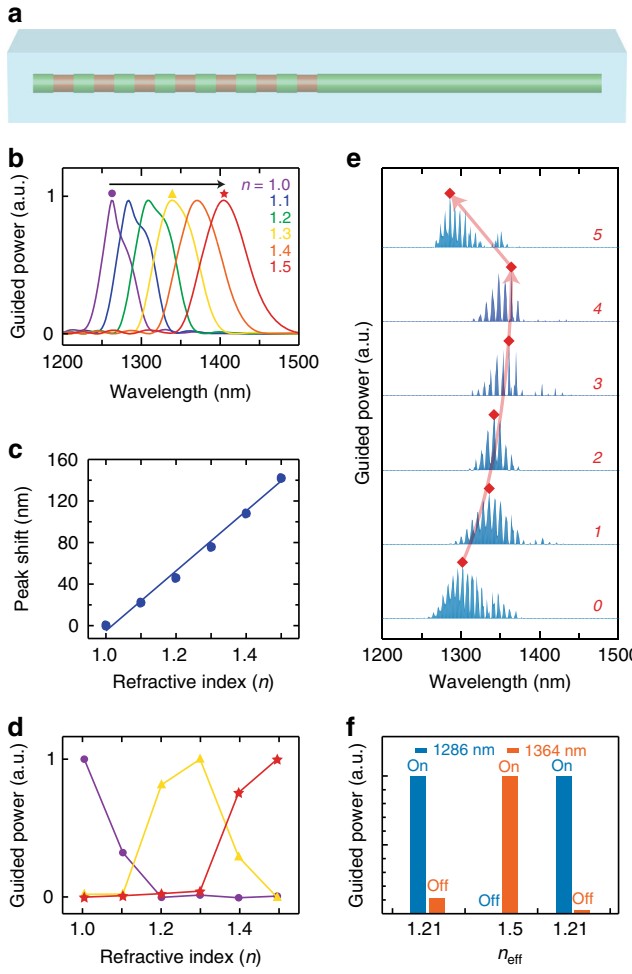

**Fig. 5** Optical switching in a GSL-WG. **a** Schematic of a GSL-WG in which the refractive index, $n$, of the surrounding medium (blue) changes from 1 to 1.5. **b** Simulated guided power spectra for a GSL-WG with $d = 280$ nm, $e = 250$ nm, and $p = 800$ nm for $n$ varying from 1 (blue) to 1.5 (red) in steps of 0.1. **c** Plot of guided peak position shift versus refractive index $n$. **d** Normalized guided power versus refractive index at wavelengths of 1262 nm (purple circles), 1338 nm (yellow triangles), and 1404 nm (red stars). Spectral positions correspond to the marker symbols in **b**. **e** Experimental guided power spectra of a NW WG with a GSL of $d = 280$ nm and $p = 820$ nm measured without PMMA (0), after sequential additions of up to 4 PMMA layers (1–4), and after removal of PMMA (5). Red diamonds and lines denote the spectral position with maximal guided power for each spectrum. **f** Normalized guided power at wavelengths of 1286 and 1364 nm derived from spectra labeled 0, 4, and 5 in **e**, which correspond to estimated effective indices of 1.21, 1.5, and 1.21, respectively

values correspond to round-trip reflectance values of 3% and 1.2% for WG$_1$ and WG$_2$, respectively. The appearance of this Fabry-Perot pattern confirms the high-quality and low loss of the GSL-WG system, and suggests a potential route for further spectral filtering of the guided light.

**Sensing and switching with a GSL-WG NW**. The guiding characteristics of the GSL-WG system are sensitive not only to geometric parameters but also to the refractive index, $n$, of the surrounding medium, as shown schematically in Fig. 5a. A change in $n$ from 1 to 1.5 in steps of 0.1 refractive index units (RIUs) results in a shift of the maximal guided power by 142 nm,

as shown by the simulations in Fig. 5b, c. This large spectral shift corresponds to a sensitivity of 270 nm RIU$^{-1}$, which is comparable to the sensitivity of planar dielectric metasurface sensors[37] and suggests potential applications of the GSL system in both sensing[37] and optical switching[40]. Figure 5d shows the progressive change in guided power as a function of $n$ at three select wavelengths, exemplifying the sensing and on/off switching characteristic that can be achieved at each wavelength by relatively small changes in $n$. However, in comparison to typical microresonators and planar metasurfaces, the finite length and nanoscale dimensions of the NW GSL cause a relatively low quality factor and wide bandwidth for the guided light. This difference highlights an inherent but common tradeoff in the design of optical components in the nanoscale regime; nevertheless, several index-changing strategies, such as the electro-optic effect[41,42] and photoisomerization[43], offer sufficiently large index modulation to enable switching in the NW GSL-WG system.

We experimentally demonstrate reversible, passive sensing, and optical switching by sequentially adding up to 4 poly(methyl methacrylate) (PMMA) layers over a NW GSL-WG on a SiO$_2$ glass substrate. Each successive PMMA layer increases the effective index of the environment. As shown by the guided spectra in Fig. 5e, the PMMA layers induce up to a 78 nm spectral shift of the maximum guided power, from 1286 to 1364 nm. This shift corresponds to a change of the effective refractive index ($n_{eff}$) of ~0.29 RIU, from ~1.21 to ~1.50, which is approximately the expected change considering the fixed $n$ of the substrate. Moreover, the spectrum reverts to its original position upon removal of all PMMA layers, thus demonstrating that the process is reversible. Figure 5f shows the relative guided power measured at wavelengths of 1286 and 1364 nm as a function of $n_{eff}$. The data exemplifies the expected sensing and optical switching behavior of the GSL-WG system, allowing the guiding of a specific wavelength to be modulated and even turned on and off by the choice of $n_{eff}$.

In conclusion, we have demonstrated that periodic geometric perturbation of a NW allows coupling of a Mie resonance and a BGS. The coupling creates a pronounced dip in scattering spectra as predicted by both numerical finite-element modeling and analytical modeling using TCMT. The periodic GSL structure acts as a Fourier frequency selector to determine the wavelength of the BGS to be coupled with the Mie resonance. The effect can be readily tuned from visible through near-infrared wavelengths by controlling geometric parameters, as proven by experimental measurements of extinction of individual GSL NWs. We demonstrate selective, Fourier-transform-limited guiding of light up to telecommunication wavelengths with near-unity coupling efficiency, highlighting the unique ability of the Mie-BGS coupling mechanism to direct energy from the strong Mie resonance light-matter interaction into a NW WG. We also demonstrate a simple optical switch that takes advantage of the spectral sensitivity of the coupled wavelength to the surrounding medium. The Mie-BGS light coupling mechanism thus offers a new platform for controlled light management when designing optical circuits.

## Methods
**Numerical modeling**. Finite-element optical simulations were performed using COMSOL Multiphysics software by running scattered field calculations with a TM-polarized plane wave or a focused Gaussian wave defined over three-dimensional NW GSL structures. For plane wave simulations, a periodic boundary condition was applied in the axial direction at each side of a GSL repeating unit, and domains to the side of the NW GSL were surrounded by air and perfectly matched layers (PMLs). For simulations with a focused Gaussian beam, NW waveguide segments on each side of the GSL were placed in PMLs. $Q_{sca}$ and $Q_{guided}$ were calculated by integrating Poynting vectors across the outer surface of a NW in all directions and

across a cross-sectional area at the end of a NW waveguide, respectively, and dividing by the optical power incident on the projected area of the GSL. Propagation to both right and left directions was taken into account for calculating $Q_{guided}$, and absorption efficiency was added to $Q_{sca}$ to calculate extinction efficiency, $Q_{ext}$.

**NW GSL synthesis**. Si NWs were grown in a home-built, hot-wall chemical vapor deposition system with silane (SiH$_4$; Voltaix), phosphine (PH$_3$; diluted to 1000 ppm in H$_2$, Voltaix), hydrogen chloride (HCl anhydrous; Matheson TriGas; 5 N research purity grade), hydrogen (H$_2$; Matheson TriGas; 5 N semiconductor grade), and with Au nanoparticle catalysts (150–300 nm; Sigma-Aldrich) of various diameters in a 1-inch quartz-tube furnace (Lindberg Blue M). All NWs were nucleated at 480 °C with 2.00 standard cubic centimeter per minute (sccm) of SiH$_4$, 4.00 sccm of HCl, and 194.0 sccm of H$_2$ at 40 Torr total reactor pressure for 30 min, which yields an average growth rate of 600 nm/min. A long section of n-type Si was grown for a desired length after nucleation by turning on the PH$_3$ flow at a fixed flow-rate ratio of 1:10 SiH$_4$:PH$_3$, and the same gas phase ratio was used throughout the study. Doping modulation for GSL profiles was performed by abruptly turning on and off the PH$_3$ flow at a constant time interval, followed by growing a long section of n-type again until growth termination. For doping-inverted GSLs, the PH$_3$ flow profile was inverted, and the final intrinsic NW sections were grown for desired WG lengths. NW GSLs were then subjected to etching in aqueous KOH solution (20 weight%) for standard doping or in buffered hydrofluoric acid (BHF, Transene BHF Improved, 2 volume%) for inverted doping as described elsewhere[36]. For optical and electron microscopy characterization, NWs were transferred after etching onto standard glass microscopy slides (Fisher Scientific) coated with ~3 nm of ITO by sputtering (Kurt Lesker PVD 75) to facilitate SEM imaging. For optical switch experiments, sequential layers of PMMA (MicroChem 950PMMA.A4) were spun cast (Laurell WS-650MZ-23NPP) at 4000 rpm and baked at 150 °C.

**Optical and microscopy characterization**. A broadband laser from a super-continuum source (NKT Photonics; SuperK Extreme EXB-6) was directed into a monochromator (Princeton Instruments; Acton SP2300) to output a single wavelength with <2 nm bandwidth. The laser was collimated into a beam size of 1 mm, polarized by a Glan-Thompson polarizer, and split evenly into reference and a probe arms for balanced detection by Nirvana balanced photoreceivers (Newport; Nirvana Auto-Balanced Photoreceiver 2007 or 2017, depending on the required wavelength). The reference beam was directly fiber-coupled to the photoreceiver, and the probe beam was directed to one side of the back aperture of a reflective objective (Thorlabs; LMM-40×-P01) to deeply under-fill the aperture and achieve quasi-plane wave illumination with a low numerical aperture (see Supplementary Fig. 9). The transmitted probe beam was re-collimated by a matched objective and fiber-coupled to the photoreceiver. The optical power was collected with the probe beam placed on and off the NW by modulating the substrate position using a piezo positioner (Mad City Labs; Nano-LP 200). Measured extinction (%) was calculated as $(1 - T) \times 100$ with $T = I/I_0$, where $I$ and $I_0$ are transmitted powers collected with the beam on and off the NW, respectively. The position modulation-extinction collection cycle was repeated twice and averaged at every wavelength. Optical imaging was performed by a visible-short wave infrared (Vis-SWIR) camera (Raptor Photonics; Ninox Vis-SWIR 640) with a halogen lamp illuminating in a dark-field configuration for locating samples or with a laser beam for collection of guided spectra. The guiding spectra were collected by taking optical images at every wavelength and integrating the signal within the NW tip region. SEM images were collected with an FEI Helios 600 Nanolab Dual Beam System, and geometrical parameters of each GSL were determined using our home-written image analysis software[30]. Duty cycles for NW GSLs used in panels a–c in Fig. 2 were calculated to be 50 ± 2%, 50 ± 2%, 49 ± 2%, respectively, indicating that the length of segments in each GSL is p/2 within ~1%.

**Data availability**. The data that support the findings of this study are available from the corresponding author upon reasonable request.

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

## Acknowledgements

This research was supported by the National Science Foundation (NSF) through Grant DMR-1555001. S.K. acknowledges a Kwanjeong Scholarship, D.J.H. acknowledges a NSF graduate research fellowships, and J.F.C. acknowledges a Packard Fellowship for Science and Engineering. H.-G.P. acknowledges support by the National Research Foundation of Korea (NRF) grant funded by the Korean government (MSIT) (Nos. 2009-0081565 and 2017R1A4A1015426). This work made use of instrumentation at the Chapel Hill Analytical and Nanofabrication Laboratory (CHANL), a member of the North Carolina Research Triangle Nanotechnology Network (RTNN), which is supported by the NSF (ECCS-1542015) as part of the National Nanotechnology Coordinated Infrastructure (NNCI). We thank Christopher W. Pinion for initial discussions on numerical calculations.

## Author contributions

S.K. and J.F.C. designed the experiments and analyzed the data; S.K. and K.-H.K. performed numerical and analytical calculations; S.K. constructed the laser microscope and performed optical experiments; D.J.H contributed to NW synthesis, instrument construction, and data analyses; H.-G.P. assisted in the data analysis and manuscript preparation; S.K. and J.F.C. wrote the paper with input from all authors; J.F.C. supervised all aspects of the project.

## Additional information

**Competing interests:** The authors declare no competing interests.

