## [Peer Review File · Nature Communications]

Reviewers' comments:

Reviewer #1 (Remarks to the Author):

Review of „Nanowire geometric superlattice optical switches“

The authors describe in their paper the coupling of Mie-resonances and bound guided states in silicon nanowires with a periodic diameter variation. The excitation of guided modes in straight sections of the nanowires as well as the sensitivity of the coupling to the external refractive index are investigated as well.

Overall the paper is very well written and offers a wealth of information on the involved modes. Clearly the authors aim to convey a deeper understanding of the coupling between the modes which lead to the characteristic dips in the scattering spectra and their parameter dependence. The figures are well prepared and very informative. The supplementary information and videos are helpful and support the understanding further. The authors pay special attention to experimental features even if they are not expected from the simple original theory (e.g. the appearing „third peak“ in the scattering spectra) - and can explain them nevertheless by considering the specific experimental conditions (Gauss-beam excitation). The described work is systematic and thorough. In this way the paper is of high interest to researchers working in the timely field of Mie-resonances, dark states and bound-states in the continua in optics and photonics.

However the authors should consider clarifying the effect of the Mie-resonance/bound state – coupling with respect to simple grating coupling to a silicon nanowire. In the moment the paper focuses only on the fashionable sounding „bound guided states“ and emphasizes the Mie-resonance, which should somehow funnel the light into the BGS. How do the scattering spectra look, when the period of the diameter variation leads to a BGS which is far off from the Mie-resonance e.g. when the period is longer shifting the lattice resonance further to the red? Then one could separate the two effects (Mie-resonance and grating coupling) clearly and first treat them separately. A simulation of this should be easily possible since the authors have already done similar things for Fig S2 and a short explanation and picture could be introduced in the supplementary materials. Afterwards the discussed specific case of Mie-superlattice coupling can be followed as the more interesting and complex scenario.

Although the authors have already put a lot of attention to the explanation of the coupling there are still some points to be clarified to avoid ambiguities and consolidate a correct understanding. The authors state on page 5 that the BGS is formed by two counterpropagating guided modes of the same frequency and wave number. It seems that the excitation of both guided modes (and therefore the creation of the BGS) is only possible for exact normal incidence from the side. A tilt of the excitation at fixed frequency would excite only one propagating guided mode (grating coupler effect) and the BGS should not form then. Is this correct? Would the sharp dip in the scattering then still exist or does that need the standing wave BGS, which is only formed by excitation of both counterpropagating modes? The authors should clarify this, so that the limitations for the formation of the BGS are clear.

In addition the BGS appears to be the same as photonic bandedge states in periodically corrugated waveguides, where the phase difference ϕ between the Ar- and the A1 mode is just 0 or π , so that once the field concentrates in the wide regions of the waveguide (ω_+) and once in the narrow regions (ω_-). While in the paper both modes (Ar- and the A1) are excited simultaneously from the side, in a photonic crystal the counterpropagating mode usually appears due to Bragg-reflection. If the authors agree with this interpretation it might be worth mentioning it at some point (in the paper or the supplementaries) so that readers who are familiar with photonics crystals can see the analogy directly.

Furthermore the authors already explain well, that the frequencies ω_+ and ω_- are actually Fano-like interferences of BGS and Mie-resonance, which are formed by a π phase difference between the two. Is this phase difference related to the ϕ -phase difference of the BGS (from page 5)? When one looks at Fig. 1d, one might be tempted to think so as the plots for ω_+ and ω_- appear just shifted. However a deeper thinking (e.g. along the lines of the photonic crystal arguments) would contradict this. The authors should therefore clarify this point.

On the other hand, the statement, that ω_+ and ω_- are formed by the interferences of BGS and Mie-resonance, might be crucial to understand why the two upper plots of the H-field in Fig. 1d only show a snapshot of positive values. Usually one would expect positive as well as negative values for a pure standing wave BGS (similar as on the bottom plot of 1d). If the positive H-field of the Mie-resonance basically overpowers the lateral positive-negative H-field variation, the authors should remark on this, to avoid suspicion in the top two plots of Fig. 1d.

Overall the discussion of the Mie-BGS-coupling forms the heart of the paper. The part about the optical switching is not so crucial, since one would expect shifts of resonances, when the refractive index changes in the surrounding. The authors should therefore also choose a title which better suits the focus of the paper.

One more technical remark: When the authors speak about simulated „scattering spectra“ – what does this actually mean? Is the extinction modelled - so basically the loss in straight-through transmission? Or is the scattered light appearing in all other directions (besides the forward direction) summed up? This should be clarified (e.g. in the methods section).

When the authors can clarify the mentioned points I will recommend the paper for publication in Nature Comm.

Reviewer #2 (Remarks to the Author):

In their manuscript, Kim et al. present silicon nanowires with a geometric superlattice along the longitudinal direction. They show through modelling with FEM and TCMT that in such structures Mie resonances can couple to bound guided states, leading to narrowband extinction from an incident beam and guiding inside the nanowire. The interpretation is sound and the overall presentation is very good, combining in an appealing way the illustration of the concepts, the simulation data and the measurement results.

However, the claim, which is also in the title, that an “optical switch” is presented seems very far-fetched. What is observed (Fig. 5) is a shift in the guided wavelength as a function of PMMA coverage. Usually, one thinks of a switch when there is a control and a signal that is switched as a function of this control. In view of an integrated component for optical communication (which actually could make more sense than optical computation that the authors put forward), it is unclear how the present nanowire device could be employed as a switch.

Still, a device which exhibits a wavelength shift as a function of the evanescently coupled cladding material coverage could be useful as a sensor, but then its performance should be compared to other optical sensors with the typical metrics, e.g. the sensitivity (nm shift per refractive index unit (RIU)). In its present form, the manuscript would be rather suitable for a more specialized journal (e.g. Optics Express), as it convincingly describes the coupling of diameter-modulated nanowires with incident beams, but I do not see the broad interest that would be required for the wider audience of Nature Communications.

Here are some additional, minor suggestions:

1. For the periodic modulation, a duty cycle of 50% is assumed, i.e. all segments have exactly the same length. How precise is this achieved in the sample fabrication? Looking at some SEM pictures, e.g. Fig. 2a, the actual fabricated structures appear not to have $p/2$ lengths everywhere and also slight variations along the wire. In general, the measurement error and statistical variation of the measured dimensions (diameters etc.) should be given to allow assessment of the fabrication precision and variations.
2. All graphs showing the extinction have no vertical axis ticks, i.e., it is not even clear if it is logarithmic or linear scale and the missing ticks and absolute numbers make it hard to allow quantitative conclusions from the graphs (depth of the dip etc.).
3. The cylindrical shape and the small transverse diameter limit intrinsically the coupling efficiency of an incident Gaussian beam. What is the (calculated) total coupling efficiency of Gaussian beam? To what degree could it be optimized using a cylindrical beam?
4. The Fabry-Perot patterns in Fig. 4f should be analyzed quantitatively in order to obtain the finesse and compare if it matches with the expected length / effective index and reflectivity.
5. Supplementary Figure S9, caption: “... of 31 and 53 mm”. Should be micrometer.

Response to Reviewer Comments:

Reviewer: 1

The authors describe in their paper the coupling of Mie-resonances and bound guided states in silicon nanowires with a periodic diameter variation. The excitation of guided modes in straight sections of the nanowires as well as the sensitivity of the coupling to the external refractive index are investigated as well.

Overall the paper is very well written and offers a wealth of information on the involved modes. Clearly the authors aim to convey a deeper understanding of the coupling between the modes which lead to the characteristic dips in the scattering spectra and their parameter dependence. The figures are well prepared and very informative. The supplementary information and videos are helpful and support the understanding further. The authors pay special attention to experimental features even if they are not expected from the simple original theory (e.g. the appearing „third peak“ in the scattering spectra) - and can explain them nevertheless by considering the specific experimental conditions (Gauss-beam excitation). The described work is systematic and thorough. In this way the paper is of high interest to researchers working in the timely field of Mie-resonances, dark states and bound-states in the continua in optics and photonics.

Author Response:

We thank the reviewer for the positive comments. We have addressed all the reviewer's concerns in detail below, and we believe these changes have significantly improved our paper.

1. However the authors should consider clarifying the effect of the Mie-resonance/bound state – coupling with respect to simple grating coupling to a silicon nanowire. In the moment the paper focuses only on the fashionable sounding „bound guided states” and emphasizes the Mie resonance, which should somehow funnel the light into the BGS. How do the scattering spectra look, when the period of the diameter variation leads to a BGS which is far off from the Mie-resonance e.g. when the period is longer shifting the lattice resonance further to the red? Then one could separate the two effects (Mie-resonance and grating coupling) clearly and first treat them separately. A simulation of this should be easily possible since the authors have already done similar things for Fig S2 and a short explanation and picture could be introduced in the supplementary materials. Afterwards the discussed specific case of Mie-superlattice coupling can be followed as the more interesting and complex scenario.

Author Response:

As the reviewer suggested, we calculated the scattering spectrum of a NW GSL with an off-resonance, long pitch ($p = 1000$ nm), and compared it with an on-resonance simulation ($p = 420$ nm) to examine the separation of BGS coupling from the Mie resonance (new Supplementary Figure S3b and d). For the long pitch, our simulation shows that the scattering dip disappears and no energy is transferred into the BGS because there is no coupling between the incident light and the BGS in the absence of a Mie resonance. Instead, a higher-order ($m = 3$) resonance by the

GSL is observed over the Mie resonance. These results demonstrate the important role of the Mie resonance for coupling energy into the BGS, and they also demonstrate the absence of a pure grating effect in the GSL structures.

Together with the new Supplementary Figure S3 and its figure caption, we have added the following sentence to the revised main text:

(Line 121-122 in page 6) “Substantially larger values of p shift ω_g off the Mie resonance envelope, causing the scattering dip to disappear and no power to be transferred to the BGS (Figure S3).”

[New Supplementary Figure S3b]

2. Although the authors have already put a lot of attention to the explanation of the coupling there are still some points to be clarified to avoid ambiguities and consolidate a correct understanding. The authors state on page 5 that the BGS is formed by two counterpropagating guided modes of the same frequency and wave number. It seems that the excitation of both guided modes (and therefore the creation of the BGS) is only possible for exact normal incidence from the side. A tilt of the excitation at fixed frequency would excite only one propagating guided mode (grating coupler effect) and the BGS should not form then. Is this correct? Would the sharp dip in the scattering then still exist or does that need the standing wave BGS, which is only formed by excitation of both counterpropagating modes? The authors should clarify this, so that the limitations for the formation of the BGS are clear.

Author Response:

As the reviewer pointed out, the excitation of two counter propagating guided modes, or an “ideal BGS” with $A_r = A_l$ (as stated in the text), is only possible under exact normal incidence because of momentum conservation and would not form upon tilt of the beam. Tilted illumination results in a splitting of the BGS dip into two sharp dips dominated by a right- and left-propagating BGS modes, as shown in our simulation (new Supplementary Figure S4). To clarify this point and the limitation for the formation of the BGS, we added the new Supplementary Figure S4 and the description of the incident angle dependence to the revised main text:

(Line 122-125 in page 6) “In addition, a tilt of incoming plane wave provides a non-zero momentum along the NW axis, causing the BGS scattering dip to split into two separate dips dominated by a right- and left-propagating BGS state (Figure S4); however, we focus on normal incidence illumination in this study.”

[New Supplementary Figure S4]

In addition, we changed the naming of the Mie resonance mode from HE_{11} to TM_{11} to clarify that we are specifically referencing the modes generated by the illumination normal to the NW axis:

(Line 110-111 in page 6): “The dominant peak in the uniform NW spectrum corresponds to the fundamental TM_{11} magnetic dipolar Mie resonance (directly related to the HE_{11} guided mode¹³) centered at ω_m .”

3. In addition the BGS appears to be the same as photonic bandedge states in periodically corrugated waveguides, where the phase difference ϕ - between the A_r - and the A_l mode is just is 0 or π , so that once the field concentrates in the wide regions of the waveguide (ω -) and once in the narrow regions (ω +). While in the paper both modes (A_r - and the A_l) are excited simultaneously from the side, in a photonic crystal the counterpropagating mode usually appears due to Bragg-reflection. If the authors agree with this interpretation it might be worth mentioning it at some point (in the paper or the supplementaries) so that readers who are familiar with photonics crystals can see the analogy directly.

Author Response:

We do not think that we can make a direct analogy between photonic crystal band edge states and the eigenmodes A_+ and A_- appearing at ω_+ and ω_- . The eigenmodes satisfy the equation $A_{\pm} = A_m \pm A_g$, which is different from photonic crystal band edge states. We explain the origin of the eigenmodes A_+ and A_- in greater detail in our response to comments 4 and 5 below. To clarify the origin of the modes, we have added to the text:

(Line 150-154 in page 7) “The eigenmodes A_+ and A_- thus represent interference between the Mie resonance and BGS, in which the axially uniform amplitude of the Mie resonance overpowers the oscillating amplitude of the BGS with a relative phase shift of π for the two eigenmodes. This result corresponds to the numerically calculated mode profiles in Figure 1d for the adjacent peaks on the blue (ω_+) and red (ω_-) sides of the scattering dip.”

4. Furthermore the authors already explain well, that the frequencies ω_+ and ω_- are actually

Fano-like interferences of BGS and Mie-resonance, which are formed by a π phase difference between the two. Is this phase difference related to the φ -phase difference of the BGS (from page 5)? When one looks at Fig. 1d, one might be tempted to think so as the plots for ω^+ and ω^- appear just shifted. However a deeper thinking (e.g. along the lines of the photonic crystal arguments) would contradict this. The authors should therefore clarify this point.

Author Response:

The π phase difference between ω^+ and ω^- modes is not related to the φ of the BGS. The π phase difference between ω^+ and ω^- modes originates from eigen modes $A_{\pm} = A_m \pm A_g$ (directly implying the π shift), while the φ term in the BGS expression was intended to provide a spatial reference point such that antinodes of a standing wave should be located in the center of each diameter segment. To clarify this point and avoid ambiguity, we modified the BGS expression to replace φ with y_0 , as follows:

(Line 93-96 in page 5) "... $A_g = [A_r(x,z)B(y)\exp(ik_{g,L}(y-y_0)-i\omega_g t) + A_l(x,z)B(y)\exp(-ik_{g,L}(y-y_0)-i\omega_g t)]$, ..., where y is the spatial position along the NW axis, y_0 is the spatial origin chosen to coincide with the center of a GSL segment, ..."

5. On the other hand, the statement, that ω^+ and ω^- are formed by the interferences of BGS and Mie resonance, might be crucial to understand why the two upper plots of the H-field in Fig. 1d only show a snapshot of positive values. Usually one would expect positive as well as negative values for a pure standing wave BGS (similar as on the bottom plot of 1d). If the positive H-field of the Mie-resonance basically overpowers the lateral positive-negative H-field variation, the authors should remark on this, to avoid suspicion in the top two plots of Fig. 1d.

Author Response:

Yes, the reviewer is correct in this statement. As the reviewer suggested, we revised the main text to clarify this point as follows:

(Line 150-154 in page 7) "The eigenmodes A_+ and A_- thus represent interference between the Mie resonance and BGS, in which the axially uniform amplitude of the Mie resonance overpowers the oscillating amplitude of the BGS with a relative phase shift of π for the two eigenmodes. This result corresponds to the numerically calculated mode profiles in Figure 1d for the adjacent peaks on the blue (ω^+) and red (ω^-) sides of the scattering dip."

6. Overall the discussion of the Mie-BGS-coupling forms the heart of the paper. The part about the optical switching is not so crucial, since one would expect shifts of resonances, when the refractive index changes in the surrounding. The authors should therefore also choose a title which better suits the focus of the paper.

Author Response:

We agree with the reviewer that the Mie-BGS coupling is the heart of the paper, but at the same

time, it enables the important physics needed for optical switching. To reflect both key points of the manuscript, we changed the title of the paper as follows: “Mie-Coupled Bound Guided States in Nanowire Geometric Superlattices”.

7. One more technical remark: When the authors speak about simulated „scattering spectra“ – what does this actually mean? Is the extinction modelled - so basically the loss in straight-through transmission? Or is the scattered light appearing in all other directions (besides the forward direction) summed up? This should be clarified (e.g. in the methods section).

Author Response:

To clarify the meaning of our simulation, we changed scattering, extinction, and guided power into scattering efficiency (Q_{sca}), extinction efficiency (Q_{ext}), and guided efficiency (Q_{guided}), respectively, in the revised figures. These values were calculated by integrating Poynting vectors across the outer surface of a NW in all directions and across a cross-sectional area at the end of a NW waveguide, respectively, and dividing by the optical power incident on the projected area of the GSL. More detailed description was shown as follows:

(Line 369-374 in page 19) “ Q_{sca} and Q_{guided} were calculated by integrating Poynting vectors across the outer surface of a NW in all directions and across a cross-sectional area at the end of a NW waveguide, respectively, and dividing by the optical power incident on the projected area of the GSL. Propagation to both right and left directions was taken into account for calculating Q_{guided} , and absorption efficiency was added to Q_{sca} to calculate extinction efficiency, Q_{ext} .”

Reviewer: 2

In their manuscript, Kim et al. present silicon nanowires with a geometric superlattice along the longitudinal direction. They show through modelling with FEM and TCMT that in such structures Mie resonances can couple to bound guided states, leading to narrowband extinction from an incident beam and guiding inside the nanowire. The interpretation is sound and the overall presentation is very good, combining in an appealing way the illustration of the concepts, the simulation data and the measurement results.

Author Response:

We thank the reviewer for the positive comments. We have addressed all the reviewer’s concerns in detail below, and we believe these changes have significantly improved our paper.

However, the claim, which is also in the title, that an “optical switch” is presented seems very far-fetched. What is observed (Fig. 5) is a shift in the guided wavelength as a function of PMMA coverage. Usually, one thinks of a switch when there is a control and a signal that is switched as a function of this control. In view of an integrated component for optical communication (which

actually could make more sense than optical computation that the authors put forward), it is unclear how the present nanowire device could be employed as a switch.

Author Response:

We thank the reviewer for raising this question regarding the term “optical switch”. In our experiment (Fig. 5), the signal (*i.e.* light guiding at a specific wavelength) is turned on and off by the control (*i.e.* refractive index of the surrounding). To highlight this switching functionality, we have added new figure panels, d and f, to the revised Figure 5. A passive switching scheme is shown in this study, but this scheme demonstrates the relevant physics needed to achieve an active switch, as mentioned at the end of the main text.

[Revised Figure 5d and f]

We have also added the following sentences to the revised main text:

(Line 329-331 in page 17) “Figure 5d shows the progressive change in guided power as a function of n at three selected wavelengths, exemplifying the on/off switching characteristic that can be achieved at each wavelength by relatively small changes in n .”

(Line 340-343 in page 18): “Figure 5f shows the relative guided power measured at wavelengths of 1286 and 1364 nm as a function of n_{eff} . The data exemplifies the expected optical switching behavior of the GSL-WG system, allowing the guiding of a specific wavelengths to be turned on and off by the choice of n .”

In addition, to partially deemphasize the switching application, we have updated the title to: “Mie-Coupled Bound Guided States in Nanowire Geometric Superlattices”.

Still, a device which exhibits a wavelength shift as a function of the evanescently coupled cladding material coverage could be useful as a sensor, but then its performance should be compared to other optical sensors with the typical metrics, e.g. the sensitivity (nm shift per refractive index unit (RIU)). In its present form, the manuscript would be rather suitable for a more specialized journal (e.g. Optics Express), as it convincingly describes the coupling of diameter-modulated nanowires with incident beams, but I do not see the broad interest that would be required for the wider audience of Nature Communications.

Author Response:

As the reviewer suggested, we added a description of the sensitivity and compared its metric

favorably to literature, in the main text regarding our sensing application:

(Line 327-329 in page 17) “This large spectral shift corresponds to a sensitivity of 270 nm RIU⁻¹, which is comparable to the sensitivity of planar dielectric metasurface sensors³⁸ and suggests potential applications of the GSL system in sensing³⁸ and optical switching.⁴¹”

However, we disagree with the reviewer that our paper does not show broad interest. We not only demonstrate the novel physics of Mie-BGS coupling in a NW-GSL system prepared entirely by an entirely bottom-up approach but also demonstrate a new optical application of this effect. We believe that our work will be appealing to a broad audience in diverse scientific communities, including photonics, physics, and nanoscience.

Here are some additional, minor suggestions:

1. For the periodic modulation, a duty cycle of 50% is assumed, i.e. all segments have exactly the same length. How precise is this achieved in the sample fabrication? Looking at some SEM pictures, e.g. Fig. 2a, the actual fabricated structures appear not to have $p/2$ lengths everywhere and also slight variations along the wire. In general, the measurement error and statistical variation of the measured dimensions (diameters etc.) should be given to allow assessment of the fabrication precision and variations.

Author Response:

We agree with the reviewer. To assess the fabrication precision and variations, the diameters and lengths of each segment of NW GSLs in Figure 2 and Figure 4 have been quantitatively re-evaluated using image-analysis software. The values with errors are now presented in the revised captions of Figures 2 and 4. In addition, we provided a quantitative evaluation of the segment lengths in the Methods section, as follows:

(Line 415-418 in page 21) “... geometrical parameters of each GSL were determined using our home-written image analysis software.³¹ Duty cycles for NW GSLs used in panels a-c in Figure 2 were calculated to be $50 \pm 2\%$, $50 \pm 2\%$, $49 \pm 2\%$, respectively, indicating that the length of segments in each GSL is $p/2$ within $\sim 1\%$.”

2. All graphs showing the extinction have no vertical axis ticks, i.e., it is not even clear if it is logarithmic or linear scale and the missing ticks and absolute numbers make it hard to allow quantitative conclusions from the graphs (depth of the dip etc.).

Author Response:

We thank the reviewer for pointing out this issue. Following the suggestion, we now present all simulated spectra in terms of optical efficiency and all measured spectra in the $1-T$ percentage scale with appropriate tick marks and values. In addition, we added have added a description of these values to the Methods section:

(Line 369-374 in page 19) “ Q_{sca} and Q_{guided} were calculated by integrating Poynting vectors across the outer surface of a NW in all directions and across a cross-sectional area at the end of a NW waveguide, respectively, and dividing by the optical power incident on the projected area of the GSL. Propagation to both right and left directions was taken into account for calculating Q_{guided} , and absorption efficiency was added to Q_{sca} to calculate extinction efficiency, Q_{ext} .”

(Line 407-409 in page 20) “Measured extinction (%) was calculated as $(1-T) \times 100$ with $T = I/I_0$, where I and I_0 are transmitted powers collected with the beam on and off the NW, respectively.”

3. The cylindrical shape and the small transverse diameter limit intrinsically the coupling efficiency of an incident Gaussian beam. What is the (calculated) total coupling efficiency of Gaussian beam? To what degree could it be optimized using a cylindrical beam?

Author Response:

To respond to the reviewer’s comment, we have calculated Q_{guided} as the guided power at the ends of the WG divided by optical power incident on the projected area of the GSL. All guided spectra are presented in the efficiency unit Q_{guided} (revised Fig. 3d and Fig. 4c-e). Our simulation results show that the guided efficiencies range from 0.5 to 1.0 depending on the choice of geometrical parameters. For detailed description, we added the following sentences to the revised manuscript:

(Line 249-254 in page 13) “Moreover, we calculate a maximum guided efficiency (Q_{guided}) of 0.82 for coupling into the WG based on the ratio of the power measured at the end of the WG to the power incident on the projected area of the GSL. Thus, the near unity Q_{guided} highlights the high efficiency with which the Mie-BGS coupling mechanism can funnel the energy associated with the strong light-matter interaction of the Mie resonance into the guided mode of a NW.”

(Line 296-297 in page 16) “Simulated guided efficiencies ranged from 0.5 to 1.0 depending on the choice of geometrical parameters.”

However, we would not expect a substantial change in Q_{guided} upon the usage of a cylindrical beam, because the NW diameter is at the sub-wavelength scale.

4. The Fabry-Perot patterns in Fig. 4f should be analyzed quantitatively in order to obtain the finesse and compare if it matches with the expected length / effective index and reflectivity.

Author Response:

As the reviewer suggested, we have added a quantitative analysis of the Fabry-Perot effect to the revised manuscript as follows:

(Line 301-305 in page 15) “WG emission was observed from the ends of both WGs, and

spectra collected from each end show free spectral ranges (FSRs) of 16.9 nm and 7.1 nm as well as finesse values of 1.58 and 1.17 from WG1 and WG2, respectively. The FSRs are in good agreement with the expected values of ~15 nm and ~6 nm based on respective WG cavity lengths, and the finesse values correspond to round-trip reflectance values of 3.0% and 1.2% for WG1 and WG2, respectively. ”

In addition, Figure S11 was edited to show an SEM of the exact NW from which the spectra in Figure 4 were collected (the SEM image of a very similar but different NW was erroneously used in the initial version).

5. Supplementary Figure S9, caption: “... of 31 and 53 mm”. Should be micrometer.

Author Response:

We thank the reviewer for correcting the typo. The units were corrected in the revised version. Also, the length of WG1 was corrected to be 21 rather than 31 microns.

REVIEWERS' COMMENTS:

Reviewer #1 (Remarks to the Author):

The authors have clarified the mentioned points. The additional simulations further enhance the understanding and illustrates the concept further.

Reading the paper carefully again I am wondering if the observed dip in scattering spectra can also be described as a Fano-resonance, which is constructed from the two mixed Mie-BGS eigenmodes A+ and A- (page 7). In the introduction (page 3) the dip is already described as a "Fano-like feature", so it might be useful to connect this term with the introduced eigenmodes. This might offer an additional perspective on the construction of the dark state so that it can be compared with the other effects mentioned in the introduction. I leave it to the authors to decide if they want to insert one or two sentences about this into the paper.

In conclusion I recommend the new improved version of the paper for publication.

Reviewer #2 (Remarks to the Author):

The authors have implemented the improvements and referees' suggestions in the revised manuscript and addressed most of the concerns of the referees satisfactorily.

In the revised version, the authors rightly deemphasized the switching aspect, but still, I would argue that, opposed to the use as sensor, the use as switch is very implausible: Typical ways how to modulate the refractive index in optical switches are thermo-optic, Pockels, Kerr or plasma dispersion based (and not by adding/removing material layers as in this manuscript), all having in common that refractive index change is only 10^{-2} or less. Although the wavelength shift in terms of nm/RIU of the device of Kim et al. is comparable to optical switches, the peak width of ~ 50 nm (Figure 5b) is by far not (they quote ref. 40 which features a resonator with 0.04 nm peak width). Hence, I would recommend focusing the discussion of the data in Figure 5 towards sensing and remove the implausible usage for switching entirely – it is an unnecessary distraction.

Apart from this minor suggestion, I recommend publication in Nature Communication.

Response to Reviewer Comments:

Reviewer #1 (Remarks to the Author):

The authors have clarified the mentioned points. The additional simulations further enhance the understanding and illustrates the concept further.

Reading the paper carefully again I am wondering if the observed dip in scattering spectra can also be described as a Fano-resonance, which is constructed from the two mixed Mie-BGS eigenmodes A+ and A- (page 7). In the introduction (page 3) the dip is already described as a "Fano-like feature", so it might be useful to connect this term with the introduced eigenmodes. This might offer an additional perspective on the construction of the dark state so that it can be compared with the other effects mentioned in the introduction. I leave it to the authors to decide if they want to insert one or two sentences about this into the paper.

In conclusion I recommend the new improved version of the paper for publication.

Author Response:

We thank the reviewer for the positive comments. We can provide an interpretation of the effect in the context of Fano resonances, and we added the following sentence in the main text:

Page 8, line 155-157: "This eigenmode analysis can also justify the assignment of the scattering dip as a Fano resonance arising from the interaction between a sharp BGS and a broad Mie background resonance."

Reviewer #2 (Remarks to the Author):

The authors have implemented the improvements and referees' suggestions in the revised manuscript and addressed most of the concerns of the referees satisfactorily.

In the revised version, the authors rightly deemphasized the switching aspect, but still, I would argue that, opposed to the use as sensor, the use as switch is very implausible: Typical ways how to modulate the refractive index in optical switches are thermo-optic, Pockels, Kerr or plasma dispersion based (and not by adding/removing material layers as in this manuscript), all having in common that refractive index change is only 10^{-2} or less. Although the wavelength shift in terms of nm/RIU of the device of Kim et al. is comparable to optical switches, the peak width of ~ 50 nm (Figure 5b) is by far not (they quote ref. 40 which features a resonator with 0.04 nm peak width). Hence, I would recommend focusing the discussion of the data in Figure 5 towards sensing and remove the implausible usage for switching entirely – it is an unnecessary distraction.

Apart from this minor suggestion, I recommend publication in Nature Communication.

Author Response:

We thank the reviewer for the positive comments. We agree that our system suffers from having

low quality factors and broad bandwidths because the NW GSL is a single dielectric nanoresonator, and we believe this is an inevitable tradeoff for achieving optical components with a nanoscale footprint. At the same time, electro-optic effect and photoisomerization of photochromic molecules are available techniques that provide index modulations as large as 1.5, which can enable switching operations with larger bandwidths.

To address the limitations of our system, we have added the following discussion in the main text:

Page 18, line 334-340: “However, in comparison to typical microresonators and planar metasurfaces, the finite length and nanoscale dimensions of the NW GSL cause a relatively low quality factor and wide bandwidth for the guided light. This difference highlights an inherent but common tradeoff in the design of optical components in the nanoscale regime; nevertheless, several index-changing strategies, such as the electro-optic effect^{41,42} and photoisomerization⁴³, offer sufficiently large index modulation to enable switching in the NW GSL-WG system.”

We have further deemphasized the switching effect by deleting the following sentences in the main text:

Page 3, deleted: “a first step toward using this system for all-optical computing technology.”

Page 19, deleted: “The ability to tune the mode through external environment promises great potential for use as an on-chip active optical switch when employing an active optical medium such as indium tin oxide (ITO)⁴¹ or photochromic molecules.⁴²”

We have also more clearly discussed the data in Figure 5 in the context of both sensing and switching by adding the sensing application at key points:

Abstract: “Using a combined GSL-WG system, we demonstrate spectrally-selective guiding and optical switching and sensing at telecommunication wavelengths, highlighting the potential to use NW GSLs for the design of on-chip optical components.”

Page 19, lines 350-351: “The data exemplifies the expected sensing and optical switching behavior of the GSL-WG system”

Page 3, lines 64-65: “In addition, we show that the coupling wavelength is highly sensitive to the local refractive index, an effect that we use to design a sensor and optical switch.”